

# Magnesium isotope fractionation processes during seafloor serpentinization and implications for serpentinite subduction

Sune G. Nielsen[1,2,3], Frieder Klein[4], Horst R. Marschall[5], Philip A. E. Pogge von Strandmann[6], Maureen Auro[1]

[1]NIRVANA Labs, Woods Hole Oceanographic Institution, 02543 Woods Hole, MA, USA
[2]Department of Geology and Geophysics, Woods Hole Oceanographic Institution, 02543 Woods Hole, MA, USA
[3]Centre de Recherches Pétrographiques et Géochimiques, CNRS, Université de Lorraine, 15 rue Notre Dame des Pauvres, 54501 Vandoeuvre lès Nancy, France.
[4]Department of Marine Chemistry and Geochemistry, Woods Hole Oceanographic Institution, 02543 Woods Hole, MA, USA
[5]Institut für Geowissenschaften, Goethe Universität Frankfurt, Frankfurt am Main, Germany
[6]Institute of Geosciences, Johannes Gutenberg University, 55122 Mainz, Germany

*Correspondence to*: Sune G. Nielsen (snielsen@whoi.edu)

**Abstract.** Studies of magnesium (Mg) isotope ratios in subduction zone lavas have revealed small, but significant offsets from the mantle value with enrichments in the heavy isotopes. However, the very high concentration of Mg in the mantle contrasts with much lower concentrations in the subducted igneous crust and oceanic sediments, making these subduction components unlikely vehicles of the Mg isotope anomalies in arc lavas. Only serpentinites, which in various proportions form part of oceanic plates, have high Mg contents comparable to fresh mantle rocks and have thus been considered a potential source of exotic Mg in the source of arc magmas.

In this study we analyzed serpentinite samples from different oceanic settings for their Mg isotopic compositions. The majority of samples are indistinguishable from the depleted mantle ($\delta^{26}$Mg = -0.24 ±0.04 ‰) irrespective of their origin. Only a small number of seafloor-weathered serpentinites are slightly enriched in the heavy isotopes (up to $\delta^{26}$Mg = -0.14 ±0.03 ‰), implying that bulk serpentinites are unlikely sources of isotopically anomalous Mg in subduction zones.

We also developed a partial-dissolution method in which 5 % acetic acid for 180 minutes was shown to fully dissolve the minerals brucite and iowaite while leaving the serpentine mineral chrysotile essentially undissolved.

Partial dissolution of 11 bulk serpentinite samples revealed a Mg isotopic composition of brucite (±iowaite) that is systematically ~0.25 ‰ heavier than that of coexisting serpentine. Thus, preferential breakdown of brucite and/or iowaite in a subducted slab prior to serpentine could preferentially release isotopically heavy Mg, which could subsequently be transported into the source region of arc magmas. Such a scenario would require brucite/iowaite breakdown to occur at pressures in excess of 3 GPa and produce fluids with very high concentrations of Mg that could be transported to arc magma source regions. Whether these conditions are met in nature has yet to be experimentally investigated.



## 1 Introduction

The release of volatiles from the subducting slab is an important process driving arc volcanism. However, how substantial amounts of water can be transported to sufficient depths prior to being released into the mantle is highly debated. Some studies argue that sediments and hydrothermally altered oceanic crust (AOC) are the primary sources of $H_2O$, as these are directly exposed to seawater at the seafloor and are naturally located at the boundary between the slab and the mantle wedge in the subduction zone (Tatsumi, 1986; Hacker, 2008). It is unclear whether hydrous minerals such as lawsonite and phengite are sufficiently abundant and stable in many slabs to carry the amounts of water required to fuel arc volcanism (e.g. Schmidt and Poli, 1998). Additionally, serpentinite has been proposed as a vehicle to transport water to a sufficient depth to eventually trigger the formation of arc magmas (e.g. Rüpke et al., 2004). Serpentinite forms during the alteration of ultramafic rocks by water at a range of temperatures and pressures (<100–600 ºC; (Lazar, 2020)). Serpentinite can form at mid-ocean ridges when mantle rocks are tectonically uplifted to crustal levels where it interacts with heated seawater, such as on slow- and ultraslow-spreading ridges in the Atlantic, Indian, and Arctic oceans (Bach and Früh-Green, 2010; Humphris and Klein, 2018). Serpentinite can also form in tectonic windows and fracture zones of fast-spreading crust and is believed to form along deep-reaching faults in oceanic trenches (Ranero et al., 2003; Mével and Stadoumi, 1996). While direct evidence for serpentinite subduction has been found in the slow spreading crust subducting underneath the Lesser Antilles (Klein et al., 2017), subduction of serpentinite in intermediate- to fast-spreading oceanic plates in the Pacific is chiefly based on geophysical inference (Ranero et al., 2003).

Geochemical evidence for serpentinite subduction remains controversial. One example of a geochemical proxy for serpentinite subduction is magnesium (Mg) isotope compositions. Much of this attention has been spurred by the discovery of $\delta^{26}Mg$ ($\delta^{26}Mg$ is the deviation of $^{26}Mg/^{24}Mg$ from that of the standard DSM-3 (Galy et al., 2003) in ‰) values slightly higher than the mantle in some arc lavas (Teng et al., 2016; Li et al., 2017). While it has been argued that these high $\delta^{26}Mg$-values may be accounted for by fractional crystallization and crustal assimilation processes (Brewer et al., 2018), others conclude that they must originate from the sub-arc mantle source region (Teng et al., 2016; Li et al., 2017; Chen et al., 2023b). In this case, the high $\delta^{26}Mg$ values have been challenging to explain via addition of sediment and AOC into arc lava source regions, because crustal rocks have much lower Mg contents than the mantle wedge (Li and Schoonmaker, 2014; Rudnick and Gao, 2003; Mcdonough and Sun, 1995). In contrast, Mg contents of peridotite are much higher and Mg is conserved during serpentinization of peridotite (Klein and Le Roux, 2020), which means that there is potential for even moderate amounts of subducted serpentinite to modify the overall Mg isotopic budget of arc lavas. Two potential processes have been argued to connect serpentinite with the Mg isotope budget of subduction zones: (1) release of fluids from the slab into the sub-arc mantle that are enriched in the heavy Mg isotopes due to the breakdown of serpentine (Teng et al., 2016) or (2) Mg isotope fractionation during serpentinization at the seafloor that could cause serpentinites to be enriched in the heavy isotopes relative to the otherwise homogenous mantle rocks (Zhao et al., 2023).



Both hypotheses provide some challenges. It has been shown in experiments at pressures (P) and temperatures (T) relevant to subduction zones that fluids released from crustal slab lithologies (sediments and AOC) carry very little Mg (Kessel et al., 2005; Spandler et al., 2007; Carter et al., 2015; Manning, 2004), possibly due to low overall Mg solubility in fluids at these T and P. Experimental investigations and theoretical models of Mg solubility in fluids expelled during serpentinite breakdown at various subduction zone conditions (1–10 GPa and 300 to 1200 ºC) predict that these can be both enriched and depleted in

70 Mg (Stalder et al., 2001) and potentially cause Mg-metasomatism of adjacent gabbroic lithologies as in the Ligurian Alps of Italy (Codillo et al., 2022). Thus, there is no consensus to what extent fluids expelled from serpentinite breakdown affect the Mg isotope budget of arc lava source regions.

In some cases, the Mg isotope composition of serpentinite display bulk-rock $\delta^{26}$Mg values slightly higher than the mantle (Zhao et al., 2023; Wang et al., 2023; Beinlich et al., 2014; Liu et al., 2017), but there is disagreement as to the processes

responsible for these heavy isotope enrichments. Some argue that seafloor weathering is the main process responsible for these values (Wang et al., 2023; Liu et al., 2017; Li et al., 2023), whereas others conclude that the serpentinization reaction itself is associated with Mg loss that causes isotope fractionation (Zhao et al., 2023). There is, hence, a need for a more comprehensive investigation that distinguishes the Mg isotope effects of primary serpentinization and secondary seafloor weathering.

In addition to serpentine, oceanic serpentinite contains up to 15 wt.% of brucite ($(Mg,Fe)(OH)_2$) (Klein et al., 2020) and

80 theoretical considerations have indicated that the Mg isotope compositions of brucite and serpentine can be substantially fractionated relative to each other, as well as relative to the fluids from which they precipitate (Wang et al., 2019; Wimpenny et al., 2014). Given that the thermodynamic stability fields of brucite and serpentine are different (Lazar, 2020), the increase in pressure and temperature during subduction could provide an additional mechanism for preferential mobilization of isotopically fractionated Mg from serpentinites in subduction zones. However, to date no study has investigated whether

brucite and serpentine in serpentinite follow predicted Mg isotope fractionation patterns during serpentinization.

Here we present Mg isotope data for a set of serpentinites from mid-ocean ridge settings and subduction-zone forearc regions revealing no bulk Mg isotope change associated with serpentinization. We also perform partial dissolution experiments, where only brucite is dissolved, which reveal that brucite is systematically higher in $\delta^{26}$Mg than coexisting serpentine.

## 2 Samples and methods

### 2.1 Sample descriptions

Serpentinites included in this study were recovered from mid-ocean ridge, off-axis, and subduction-zone settings. Samples recovered during Ocean Drilling Program (ODP) Legs 153 (Site 920B) and 209 (Sites 1268A, 1271B, 1272A, and 1274A) on the slow-spreading Mid-Atlantic Ridge include partly to completely serpentinized harzburgite and dunite. Their secondary mineral assemblages are dominated by serpentine, brucite, and magnetite in addition to minor chlorite and iowaite (Klein et

al., 2020), and traces of Ni-bearing sulfides and alloys. Brucite, which formed together with serpentine at the expense of olivine in mesh texture, is partially and locally completely replaced by iowaite.



Samples from Deep Sea Drilling Program Leg 82 (Site 558)) were recovered off the Mid-Atlantic Ridge in 34.7 Ma crust (37°46.24' N, 37°20.61' W). Harzburgite is completely serpentinized and mainly composed of serpentine, iowaite, and maghemite. In contrast to samples from ODP Legs 153 and 209, iowaite preferentially occurs in late-stage veins that cut across mesh and bastite texture.

Samples from the Puerto Rico Trench were recovered during Cruise 19 of RV Chain (Bowin *et al.*, 1966; Klein *et al.*, 2017). Two dredges from the North Wall of the Puerto Rico Trench, which is currently undergoing subduction beneath the Caribbean Plate, include highly weathered serpentinized peridotite (dredge D10) composed of lizardite, chrysotile, minor antigorite, chlorite, hematite, goethite, clay minerals, and quartz, and Si-metasomatized serpentinite (dredge D2) chiefly composed of antigorite and talc. Dating of zircon in mafic veins that cut sample D10-9 yielded an age of 114.8 Ma, indicating that these serpentinites were originally formed at the Cretaceous Mid-Atlantic Ridge and were subsequently transported by seafloor spreading to their current location (Klein et al., 2017).

Samples from the Mariana forearc were drilled during ODP Leg 125 and 195 from Conical Seamount and South Chamorro Seamount, respectively (D'antonio and Kristensen, 2004; Fryer et al., 1992). These samples are partly to completely serpentinized peridotites chiefly composed of serpentine (lizardite, chrysotile, antigorite), brucite, iowaite, chlorite, and magnetite (Kahl et al., 2015; Klein et al., 2020). In contrast to other samples included in this study, peridotite from South Chamorro and Conical was serpentinized in the shallow mantle wedge by sediment-derived pore fluids evolved during subduction of the Pacific plate (Kahl et al., 2015; Nielsen et al., 2015; Debret et al., 2019).

## 2.2 Partial dissolution experiments

In order to investigate whether brucite, iowaite and serpentine are characterized by different Mg isotope compositions, we developed a partial dissolution protocol designed to preferentially dissolve brucite and iowaite, whereas serpentine was left in the residue. The isotope composition of serpentine could thereby be calculated via mass balance using the bulk Mg isotope composition and concentration.

We performed dissolution experiments of pure single mineral materials of brucite, iowaite and chrysotile using acetic acid to determine the kinetics of their dissolution at different acetic acid strengths. For brucite and iowaite we conducted 4 separate experiments using 5 %, 10 %, 20 % and 40 % acetic acid at ambient room temperature (~20°C), respectively. Sample aliquots were taken out after 1 min, 10 min, 100 min, 360 min (=6 h) and 1440 min (=24 h) and the Mg concentrations were determined (Fig. 1). If less than 100 % of the mineral had dissolved, then a mineral sample aliquot was also taken for Mg isotopic analysis. For chrysotile only one experiment was performed at 5 % acetic acid strength with samples taken at 80 min and 1200 min. The Mg isotope compositions were not determined for this experiment as only a few percent of the serpentine had dissolved during the experiment (Fig. 1).

We selected 11 serpentinite samples for partial dissolution. Samples were selected based on their brucite and iowaite contents, lack of extensive seafloor weathering (Klein et al., 2020), as well as being completely or almost completely serpentinized.





Thus, no or only trivial amounts of primary Mg bearing minerals (i.e. olivine, pyroxenes) remained, which allowed us to use

the bulk Mg isotope compositions to reconstruct the Mg isotope composition of serpentine in these samples via mass balance.

Concentrations of Na, Mg, Ca, Fe, Al, Cr, Mn, Co, Ni, and Zn in the partial dissolution samples were determined on a Thermo

iCap quadropole inductively couple plasma mass spectrometer (ICP-MS) located in the WHOI Plasma Facility. Concentrations

were determined via reference to serially diluted USGS reference materials AGV-2 and BHVO-2 run at the beginning and end

of each sequence. Instrumental drift was corrected by monitoring the signal of indium that was added to each sample and

reference material in a known quantity. Following previous studies from our group that use identical methods for concentration

determination we infer that the external reproducibility of these analyses are ~7 % (1sd) (Shu et al., 2022; Wang et al., 2022).

**Figure 1: Fraction of Mg dissolved in single mineral dissolution experiments. Acetic acid strength is color coded. The fraction of total Mg dissolved for brucite was normalized to the values for the end of the experiment since no solids remained, whereas chrysotile**

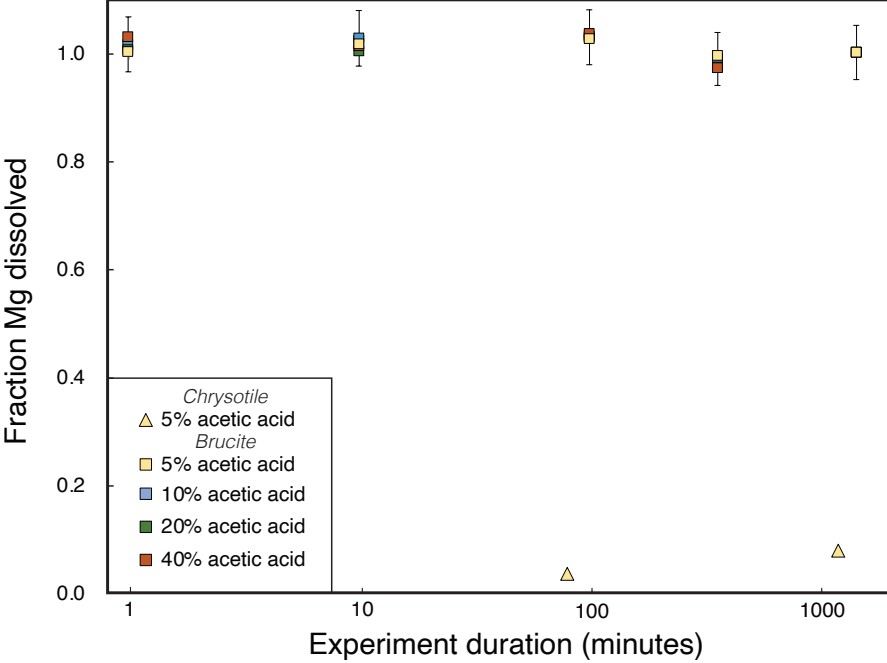

**was calculated by assuming an idealized stoichiometric formula of $Mg_3(Si_2O_5)(OH)_4$. Uncertainties for the chrysotile experiments are smaller than symbol sizes.**

## 2.3 Magnesium separation and isotope measurements

Purification of Mg from sample matrix was performed as previously detailed (Chen et al., 2023a; Pogge Von Strandmann et

al., 2011). Briefly, the dissolved samples were purified using a two-stage cation exchange column procedure, using AG50W

X-12 ion exchange resin, eluting the Mg using 2M $HNO_3$.

Magnesium isotope compositions were measured using a Nu Plasma 3 MC-ICP-MS at the LOGIC laboratories in London,

using a sample-standard bracketing method relative to DSM-3. Based on the repeated purification and analysis of reference





materials USGS BCR-2 ($\delta^{26}$Mg = -0.24 ±0.08 ‰, 2sd, n = 11), JP-1 (-0.23 ±0.07 ‰, 2sd, n = 3), and seawater ($\delta^{26}$Mg = -0.82

±0.04 ‰, 2sd, n = 15), the accuracy of our measurements is indistinguishable from other studies (Foster et al., 2010; Pogge

Von Strandmann et al., 2011), and the long-term external precision on $\delta^{26}$Mg is ±0.08 ‰ (2sd).

## 3 Results and discussion

### 3.1 Partial dissolution experiments

The single-mineral dissolution experiments reveal that both brucite and iowaite dissolve rapidly in all acetic acid

concentrations (Figs. 1 and 2). Brucite dissolved in less than 1 minute in all experiments, whereas iowaite required more than

100 minutes to fully dissolve in 5 % acetic acid. Chrysotile did not dissolve appreciably over the 1200 minutes we conducted

the experiment in 5 % acetic acid (Fig. 1). The experiments where iowaite had partially dissolved exhibit a significant Mg

isotopic difference as a function of the fraction of Mg dissolved (Fig. 2), suggesting that partial dissolution of iowaite is

associated with Mg isotope fractionation, whereby the light isotope is preferentially released into solution first. However, we

infer that the net Mg isotope fractionation factor quickly approaches 0 as a larger fraction of iowaite is dissolved. Based on

these results we chose to conduct the partial dissolution of the bulk serpentinites for 180 minutes in 5 % acetic acid. This

configuration likely ensures full dissolution of brucite and iowaite, whereas dissolution of serpentine would be minimal. The

single mineral dissolution experiments imply that it is not possible to chemically separate brucite and iowaite and, therefore,

the partial dissolution of bulk serpentinites provides information about the Mg isotope difference between brucite/iowaite and

serpentine.

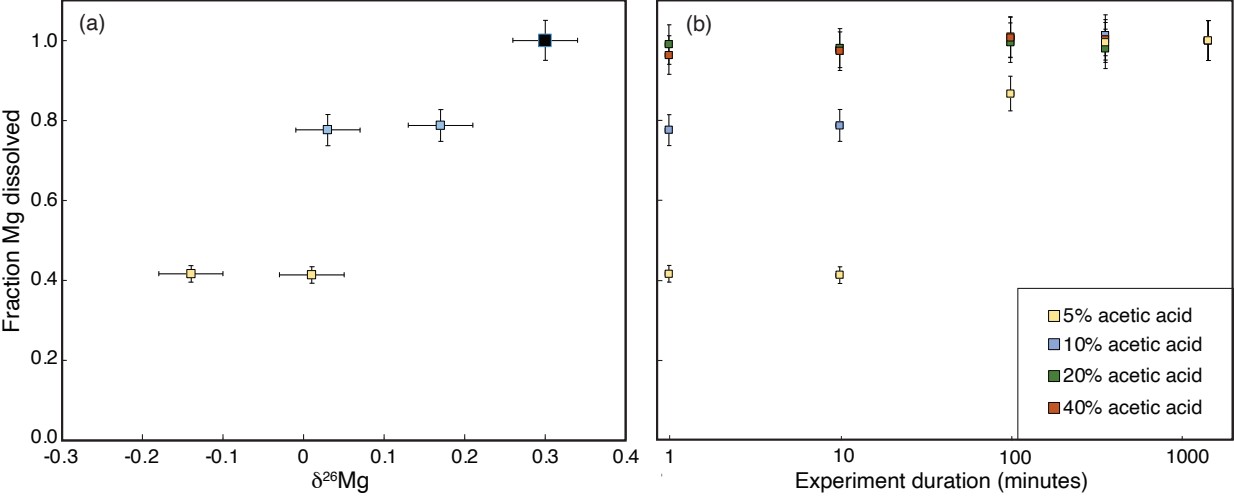

**Figure 2: Results of iowaite single mineral partial dissolution experiments. The fraction of total Mg dissolved was normalized to the values for the end of the experiment since no solids remained. Experiments without full mineral dissolution in 5 % and 10 % acetic acid were analyzed for Mg isotopes (a), whereas the fully dissolved mineral was measured once (black symbol in panel a).**



## 3.2 Bulk serpentinite data

Magnesium isotope compositions of 33 bulk serpentinite samples (Table 1) reveal only very minor variation, with the majority of samples exhibiting values within uncertainty of the mantle value (Figure 3). Only a few of the dredged serpentinite samples (those from the Puerto Rico trench) exhibit values that extend up to $\delta^{26}Mg$ = -0.14 ±0.03 ‰, slightly higher than the mantle ($\delta^{26}Mg$ = -0.24 ±0.04 ‰, (Hin et al., 2017; Teng et al., 2010)).

As previously shown based on thermogravimetric analysis, mineralogical constraints, and major element data, dredged serpentinites typically record various degrees of seafloor weathering (Klein et al., 2020), which has been argued to result in Mg isotope fractionation towards more positive $\delta^{26}Mg$-values (Wang et al., 2023; Liu et al., 2017). The serpentinite samples that were not affected by weathering, all of which were recovered by scientific ocean drilling, record no Mg isotope deviations from the mantle, which is consistent with the conservation of Mg during serpentinization (Klein et al., 2020; Klein and Le Roux, 2020). Magnesium loss and Mg isotope fractionation during serpentinization, which has been argued for in the recent literature (Zhao et al., 2023), was possibly caused by (seafloor) weathering prior to metamorphic re-equilibration of these tectonically exhumed rocks or by metasomatism. The fact that these rocks contain negligible brucite (Zhao et al., 2023) is also a strong indication that they were subject to post-serpentinization modifications, e.g., via seafloor weathering or Si-metasomatism.

Table 1: Bulk Mg isotope compositions of serpentinites

| Sample name | Location | Sampling method | $\delta^{25}Mg$ (‰) | 2se | $\delta^{26}Mg$ (‰) | 2se | n |
|---|---|---|---|---|---|---|---|
| D2-1 | Puerto Rico trench | dredge | -0.12 | 0.02 | -0.24 | 0.03 | 1 |
| D2-2 | Puerto Rico trench | dredge | -0.09 | 0.02 | -0.17 | 0.02 | 2 |
| D2-4 | Puerto Rico trench | dredge | -0.09 | 0.03 | -0.17 | 0.04 | 1 |
| D2-5 | Puerto Rico trench | dredge | -0.13 | 0.02 | -0.26 | 0.06 | 2 |
| D2-6 | Puerto Rico trench | dredge | -0.11 | 0.04 | -0.22 | 0.03 | 2 |
| D10-1 | Puerto Rico trench | dredge | -0.11 | 0.01 | -0.24 | 0.04 | 1 |
| D10-3 | Puerto Rico trench | dredge | -0.13 | 0.03 | -0.25 | 0.04 | 1 |
| D10-7 | Puerto Rico trench | dredge | -0.11 | 0.02 | -0.23 | 0.03 | 1 |
| D10-8 | Puerto Rico trench | dredge | -0.08 | 0.04 | -0.14 | 0.03 | 2 |
| D10-9 | Puerto Rico trench | dredge | -0.07 | 0.03 | -0.19 | 0.03 | 1 |
| D10-12 | Puerto Rico trench | dredge | -0.08 | 0.01 | -0.17 | 0.03 | 1 |
| D10-15 | Puerto Rico trench | dredge | -0.10 | 0.03 | -0.15 | 0.04 | 1 |
| D10-16 | Puerto Rico trench | dredge | -0.11 | 0.02 | -0.22 | 0.02 | 1 |
| 209-1268A-8R1-28-35 | 15°20FZ | drilling | -0.14 | 0.01 | -0.29 | 0.02 | 1 |
| 209-1272A-14R1-43-53 | 15°20FZ | drilling | -0.13 | 0.04 | -0.24 | 0.04 | 3 |



| | | | | | | | |
|---|---|---|---|---|---|---|---|
| 209-1268A-4R1-44-55 | 15°20FZ | drilling | -0.11 | 0.01 | -0.24 | 0.03 | 1 |
| 209-1271B-17R1-61-69 | 15°20FZ | drilling | -0.13 | 0.04 | -0.24 | 0.05 | 2 |
| 209-1272A-21R1-88-100 | 15°20FZ | drilling | -0.11 | 0.01 | -0.27 | 0.03 | 1 |
| 209-1272A-27R2-78-88 | 15°20FZ | drilling | -0.10 | 0.02 | -0.21 | 0.03 | 1 |
| 209-1274A-16R2-26-38 | 15°20FZ | drilling | -0.15 | 0.02 | -0.26 | 0.02 | 1 |
| | | | | | | | |
| 195-1200A-17R2-76-79 | South Chamorro Seamount | drilling | -0.13 | 0.03 | -0.28 | 0.02 | 1 |
| 195-1200A-11R1-47-49 | South Chamorro Seamount | drilling | -0.12 | 0.04 | -0.25 | 0.02 | 1 |
| 195-1200A-13R1-121-124 | South Chamorro Seamount | drilling | -0.08 | 0.03 | -0.18 | 0.03 | 1 |
| 125-779A-10R2-51-53 | Conical seamount | drilling | -0.12 | 0.02 | -0.28 | 0.04 | 1 |
| 125-779A-17R4-32-34 | Conical seamount | drilling | -0.17 | 0.03 | -0.26 | 0.04 | 1 |
| 125-779A-31R2-85-87 | Conical seamount | drilling | -0.12 | 0.03 | -0.25 | 0.02 | 1 |
| | | | | | | | |
| 82-558Z-42R1-9-11 | Azores off axis | drilling | -0.10 | 0.01 | -0.22 | 0.03 | 1 |
| 82-558Z-42R1-133-135 | Azores off axis | drilling | -0.12 | 0.02 | -0.23 | 0.04 | 1 |
| 82-558Z-48R1-45-46 | Azores off axis | drilling | -0.10 | 0.03 | -0.25 | 0.03 | 1 |
| | | | | | | | |
| 153-920B-2R1-80-82 | MARK | drilling | -0.11 | 0.03 | -0.26 | 0.03 | 1 |
| 153-920B-5R2-35-38 | MARK | drilling | -0.06 | 0.02 | -0.18 | 0.03 | 1 |
| 153-920B-10R1-82-86 | MARK | drilling | -0.09 | 0.04 | -0.20 | 0.04 | 1 |
| 153-920B-12R2-140-143 | MARK | drilling | -0.15 | 0.02 | -0.24 | 0.04 | 1 |

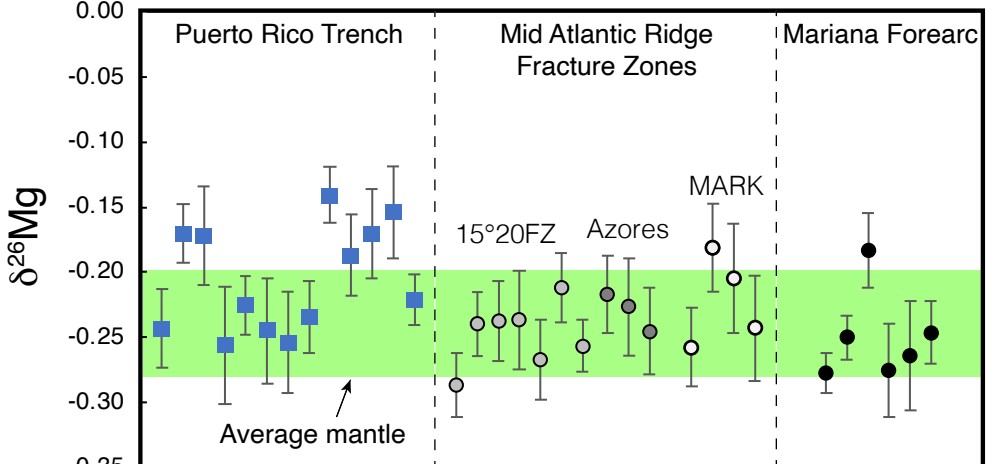



**Figure 3: Bulk-rock Mg isotopic compositions of 33 seafloor serpentinites. Error bars are 2se. Circle symbols are drill cores, whereas squares are dredged (i.e. Puerto Rico trench). The average mantle value is based on literature data (Teng et al., 2016; Hin et al., 2017).**

The Mg isotopic compositions are quasi invariant regardless of the tectonic environment that the serpentinization took place in (i.e. mid-ocean ridge, subduction zone forearc), which is consistent with previous results from similar environments (Wang et al., 2023; Liu et al., 2017). This result implies that the serpentinizing fluid composition, i.e. seawater or fluids sourced from dehydration of sediments (Nielsen et al., 2015), or the olvine-to-orthopyroxene ratio of the protolith, do not affect the Mg isotope budget of serpentinite (Pogge Von Strandmann et al., 2015). Hence, serpentinite subduction does not introduce

isotopically anomalous Mg into subduction zones and serpentinite is unlikely to form an endmember bulk source of the positive $\delta^{26}$Mg-values observed in some arcs (Teng et al., 2016; Li et al., 2017). Should these positive $\delta^{26}$Mg-values in arcs be generated by processes involving serpentinite, then these signatures must be generated in the subduction zone itself by partial mobilization of Mg from serpentinite associated with Mg isotope fractionation. Two different scenarios could achieve this: (1) either material with low $\delta^{26}$Mg-values is removed from serpentinite at relatively low pressures and temperatures, leaving higher

$\delta^{26}$Mg values in the residue that could then contribute to the arc-lava source regions; or (2) material with positive $\delta^{26}$Mg-values is released from serpentinite and is transported into the source of arc magmas (Teng et al., 2016).

One potentially critical parameter for these scenarios could be, if the dominant Mg-bearing minerals in serpentinite, i.e. serpentine and brucite, were associated with different Mg isotope compositions. This has indeed been suggested based on theoretical calculations and experiments (Wang et al., 2019; Wimpenny et al., 2014; Wimpenny et al., 2010; Gao et al., 2018).

However, the exact isotope fractionation between these two phases is not agreed upon with evidence pointing towards brucite exhibiting both higher and lower $\delta^{26}$Mg-values than serpentine at equilibrium.

### 3.3. Mg isotope fractionation between major serpentinization minerals

Of the 11 samples for which acetic-acid partial dissolution was performed, all recorded $\delta^{26}$Mg values are within uncertainty of

or higher than the bulk measurement (Table 2). We confirmed through thermogravimetric analysis (TGA) of 8 samples that complete dissolution of brucite and iowaite occurred, while serpentine was retained (Fig. S1).

**Table 2: Elemental concentrations and Mg isotope compositions of leached serpentinites**

| Sample | Mg | Fe | Al | Ca | Cr | Mn | Co | Ni | Zn | $\delta^{26}$Mg | 2se |
|---|---|---|---|---|---|---|---|---|---|---|---|
| 209-1272A-27R2-78-88 | 6.30 | 1.62 | 0.047 | 0.009 | 246 | 379 | 30 | 622 | 14 | -0.22 | 0.03 |
| 209-1272A-21R1-88-100 | 4.55 | 0.95 | 0.064 | 0.004 | 332 | 283 | 33 | 826 | 8 | -0.01 | 0.02 |
| 209-1274A-16R2-26-28 | 6.19 | 1.70 | 0.033 | 0.017 | 188 | 413 | 20 | 696 | 17 | -0.06 | 0.05 |
| 209-1271B 17R1 61-69 | 6.53 | 0.69 | 0.009 | 0.004 | 86 | 440 | 21 | 1365 | 10 | -0.04 | 0.02 |
| 82-558Z-48R1-45-46 | 4.96 | 1.77 | 0.007 | 0.012 | 25 | 233 | 83 | 2143 | 19 | -0.26 | 0.02 |
| 82-558Z-42R1-133-135 | 3.80 | 1.09 | 0.042 | 0.011 | 292 | 172 | 83 | 1795 | 11 | -0.06 | 0.03 |
| 195-1200A-11R1-47-49 | 5.35 | 1.22 | 0.009 | 0.177 | 68 | 375 | 54 | 1068 | 18 | -0.10 | 0.03 |



| | | | | | | | | | | | |
|---|---|---|---|---|---|---|---|---|---|---|---|
| 195-1200A-13R1-121-124 | 6.61 | 1.65 | 0.028 | 0.200 | 59 | 456 | 32 | 521 | 20 | -0.07 | 0.01 |
| 195-1200A-17G-76-79 | 4.71 | 1.26 | 0.018 | 0.127 | 66 | 316 | 47 | 793 | 12 | -0.10 | 0.03 |
| 153-920B-12R2-140-143 | 6.26 | 0.63 | 0.059 | 0.006 | 26 | 468 | 26 | 1424 | 9 | -0.13 | 0.02 |
| 125-779A-10R2-51-53 | 3.76 | 1.24 | 0.001 | 0.050 | 10 | 213 | 29 | 729 | 10 | -0.16 | 0.03 |

**Major elements (Mg, Fe, Al, Ca) in weight %, all others in µg/g**

Although the procedure dissolves both brucite and iowaite while leaving serpentine essentially undissolved, we note that iowaite in mesh texture is generally thought to form from replacement of brucite (D'antonio and Kristensen, 2004; Klein et al., 2020). Given that the cation stoichiometry between these two minerals is essentially the same (molar Mg/(Mg+Fe) ≈ 0.8), we infer that all the Mg included in iowaite likely was initially present in brucite. Therefore, the majority of Mg in the leachates

reflects the Mg contained in brucite during initial serpentinization. The single-mineral dissolution experiments reveal that minor amounts of Mg from serpentine (~4 %) does dissolve within the first hours of reacting 5 % acetic acid with chrysotile, whereas only little additional Mg was released when extending the experiment to 20 hours (Fig. 1). Assuming that ~4 % of serpentine-bound Mg dissolved during our bulk serpentinite leaching experiments, we calculate that 12–22 % of the Mg released during the experiments originated from serpentine (i.e. $f\mathrm{Mg_{serp,leach}}$, Table 3). Similarly, we use the bulk serpentinite

and total leached Mg contents of the samples, while assuming that all Mg in the samples are either located in serpentine, iowaite, or brucite, to calculate the fraction of Mg in the bulk sample accommodated by serpentine (i.e. $f\mathrm{Mg_{serp,bulk}}$). There is no significant correlation between the fraction of Mg released from serpentine and the Mg isotope composition of the leach (Fig. 4), suggesting that partial dissolution of serpentine did not cause substantial isotope fractionation. In any case, such a process would likely be associated with kinetic isotope fractionation similar to what was observed for partial iowaite

dissolution. That would result in lower $\delta^{26}\mathrm{Mg}$-values in the leachate, opposite to the values higher than the bulk recorded in the leach experiments (Table 2).

Because all samples leached were fully serpentinized without any remaining olivine or pyroxene, we can use mass balance to calculate the Mg isotope compositions of the major Mg bearing serpentinization minerals. We use the following two mass balance equations to determine the Mg isotope compositions of the two phases in the serpentinites (i.e. $\delta^{26}\mathrm{Mg_{serp}}$ and $\delta^{26}\mathrm{Mg_{b+i}}$):

$$f\mathrm{Mg_{serp,bulk}} * \delta^{26}\mathrm{Mg_{serp}} + f\mathrm{Mg_{b+i,bulk}} * \delta^{26}\mathrm{Mg_{b+i}} = \delta^{26}\mathrm{Mg_{bulk}} \qquad \text{(eq. 1)}$$

$$f\mathrm{Mg_{serp,leach}} * \delta^{26}\mathrm{Mg_{serp}} + f\mathrm{Mg_{b+i,leach}} * \delta^{26}\mathrm{Mg_{b+i}} = \delta^{26}\mathrm{Mg_{leach}} \qquad \text{(eq. 2)}$$

Equation 2 assumes that no Mg isotope fractionation occurred during leaching, which is the case for brucite and iowaite, since they were both fully dissolved. It is unclear if the Mg isotope fractionation observed for iowaite can be translated to serpentine or indeed whether any substantial effect would be present after 3 hours of reaction (as opposed to the 1–10 minutes where we observed small effects). The calculated Mg isotope compositions of the mineral phases (Table 3) allows an assessment of the net Mg isotope fractionation ($\Delta^{26}\mathrm{Mg}$) between brucite/iowaite and serpentine (Table 3).





**Table 3: Mg isotope mass balance of leached serpentinites**

| Sample | Mg$_{bulk}$* | Mg$_{residue}$ | Mg$_{serp,bulk}$ (4% diss.) | $f$Mg$_{serp,bulk}$ | $f$Mg$_{b+i,bulk}$ | $f$Mg$_{serp,leach}$ | $f$Mg$_{b+i,leach}$ | $\delta^{26}$Mg$_{serp}$ | $\delta^{26}$Mg$_{b+i}$ | $\Delta^{26}$Mg |
|---|---|---|---|---|---|---|---|---|---|---|
| 82-558Z-48R1, 45-46 | 22.98 | 18.01 | 18.76 | 0.82 | 0.18 | 0.151 | 0.849 | -0.24 | -0.26 | -0.02 |
| 82-558Z-42R1, 133-135 | 23.84 | 20.04 | 20.87 | 0.88 | 0.12 | 0.220 | 0.780 | -0.26 | 0.00 | +0.25 |
| 125-779A-10R-2, 51-53 | 24.13 | 20.37 | 21.22 | 0.88 | 0.12 | 0.225 | 0.775 | -0.30 | -0.13 | +0.17 |
| 153-920B-12R2, 140-143 | 24.33 | 18.07 | 18.83 | 0.77 | 0.23 | 0.120 | 0.880 | -0.28 | -0.11 | +0.17 |
| 195-1200A-11R1 47-49 | 24.67 | 19.32 | 20.12 | 0.82 | 0.18 | 0.150 | 0.850 | -0.29 | -0.06 | +0.23 |
| 195-1200A-13R1 121-124 | 25.06 | 18.45 | 19.22 | 0.77 | 0.23 | 0.116 | 0.884 | -0.22 | -0.05 | +0.17 |
| 195-1200A-17G 76-79 | 24.64 | 19.93 | 20.76 | 0.84 | 0.16 | 0.177 | 0.823 | -0.32 | -0.05 | +0.27 |
| 209-1271B 17R1 61-69 | 24.69 | 18.16 | 18.92 | 0.77 | 0.23 | 0.116 | 0.884 | -0.31 | -0.01 | +0.30 |
| 209-1272A-21R1 88-100 | 23.99 | 19.44 | 20.25 | 0.84 | 0.16 | 0.178 | 0.822 | -0.33 | 0.06 | +0.38 |
| 209-1272A-27R2 78-88 | 23.85 | 17.55 | 18.28 | 0.77 | 0.23 | 0.116 | 0.884 | -0.21 | -0.22 | -0.01 |
| 209-1274A-16R2 26-28 | 24.77 | 18.58 | 19.36 | 0.78 | 0.22 | 0.125 | 0.875 | -0.32 | -0.02 | +0.31 |

**Mg concentrations (bulk, residue and serpentine) in weight %. Mg$_{residue}$ calculated from leached Mg in Table 2**

**\* - bulk Mg concentrations previously published (Paulick et al., 2006; Klein et al., 2017)**

Although most samples contained iowaite (Supplement), one sample (82-558Z-48R1, 45-46) presented iowaite in
monomineralic veins that cut across pseudomorphic textures and likely formed via co-precipitation from mixing of seawater
and a more alkaline solution. This sample is one of the two with $\Delta^{26}$Mg < 0. We, therefore, conclude that late-stage formation
of iowaite veins likely caused precipitation of isotopically light Mg relative to what initially precipitated as brucite during
serpentinization. Unfortunately, we do not have a thin section for the other sample with $\Delta^{26}$Mg < 0 (209-1272A-27R2 78-88)
to corroborate this conclusion, but we infer that this sample also contains late stage iowaite veins.

Apart from the samples with late-stage iowaite veins, all samples reveal systematic and similar Mg isotope fractionation
between serpentine and brucite of $\Delta^{26}$Mg = +0.25 ± 0.15 ‰ (2sd), implying that brucite formed during serpentinization is
isotopically about 0.25 ‰ heavier than co-existing serpentine. This result contrasts $\Delta^{26}$Mg ~ -0.1 to -0.8 inferred for brucite
dissolution during post serpentinization alteration (Li et al., 2023) and is in stark contrast to calculations of Mg isotope
equilibrium between brucite and the serpentine mineral lizardite that imply an equilibrium isotope fractionation at 300 ˚C of
270 $\Delta^{26}$Mg ~ -1 ‰ (Wang et al., 2019). On the other hand, brucite-precipitation experiments have revealed that brucite
preferentially incorporates heavy Mg isotopes relative to the solution from which it precipitates at room temperature, with
$\Delta^{26}$Mg ~ +0.5 ‰ (Wimpenny et al., 2014). Given that brucite in the examined serpentinites likely precipitated at temperatures
of 150–300˚C (Klein et al., 2014) and that isotope fractionation factors (equilibrium and/or kinetic) roughly scale somewhere
between 1/T and 1/T$^2$ (Chacko et al., 2001; Bigeleisen and Mayer, 1947), it is expected that fractionation factors are about one
quarter (for 1/T$^2$ at 300˚C) to three quarters (for 1/T at 150˚C) of those found at room temperature. This range would correspond
to $\Delta^{26}$Mg = +0.13 to +0.38 ‰, identical to what we find here empirically (Table 3). Our data, therefore, imply that brucite
formation proceeds through precipitation from a fluid that sourced its Mg from dissolution of olivine (or previously formed
serpentine since these are isotopically very similar). Since most Mg during serpentinization forms serpentine, it is expected





that serpentine broadly retains the Mg isotope composition of the bulk starting material. On the contrary, brucite should express

the Mg isotope fractionation occurring during precipitation, since only a small fraction of the Mg in solution would likely be

used to form brucite. These observations imply that brucite and serpentine in serpentinite do not form at thermodynamic Mg

isotope equilibrium. Given that brucite and serpentine often form intergrowths suggesting simultaneous formation (Klein et

al., 2020; Hostetler et al., 1966), we infer that both minerals precipitate from the same fluid where any Mg isotope fractionation

occurring during serpentine precipitation is attenuated due to almost quantitative uptake of Mg from the solution.

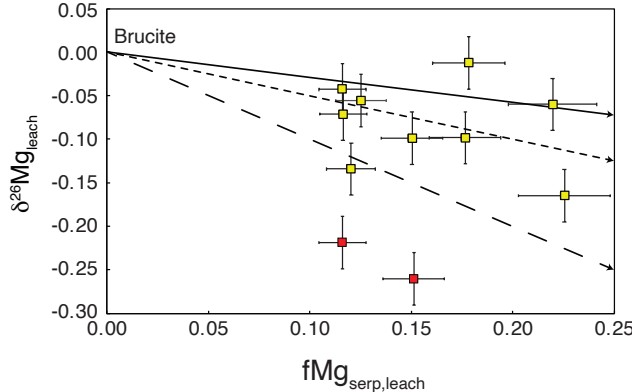

**Figure 4: Fraction of total Mg leached that originated from serpentine ($fMg_{serp,leach}$) plotted against measured Mg isotope composition of leach. Samples in dark red are those in which monomineralic veins of iowaite were identified. Arrows indicate mixing between brucite (with an assumed $\delta^{26}Mg = 0$‰) and partially dissolved serpentine (with an assumed $\delta^{26}Mg = -0.29$‰). The three lines denote congruent serpentine dissolution (full line), dissolution isotope fractionation of -0.21‰ (short-dashed line), and**

**dissolution isotope fractionation of -0.71‰ (long-dashed line).**

### 3.4. Implications for subduction zone cycling of Mg isotopes

As outlined in the introduction, several volcanic arcs have been investigated that exhibit small, but measurable positive Mg

isotope anomalies relative to that of the upper mantle, which has been inferred to be related to subduction of isotopically heavy

serpentinite (Teng et al., 2016; Li et al., 2017). However, our data for unweathered serpentinite, together with several other

data sets (Wang et al., 2023; Liu et al., 2017){Li, 2023 #5724}, reveal that bulk serpentinite is unfractionated relative to the

mantle. This demonstrates that the bulk Mg content of serpentinite cannot be responsible for the positive Mg isotope anomalies

observed in some volcanic arcs. However, given that the mineral brucite exhibits on average $\delta^{26}Mg = -0.04$ ‰ (excluding the

iowaite vein samples, Table 3), significantly higher than those observed in most arc lavas ($\delta^{26}Mg < -0.1$ ‰), preferential

mobilization of brucite-bound Mg from the slab could be responsible for the observed Mg isotope anomalies in some arc

magmas. If such a process was to operate, then breakdown of brucite would need to form a fluid rich in Mg that could operate

in three distinct scenarios of slab-to-mantle transport:

(1) migrate into the mantle wedge and cause melting in the wedge peridotite – in this case the brucite breakdown would have

to occur at ≥4 GPa, such that the fluids can be transported into the sub-arc region.



(2) migrate into the eclogite or metasediment layer and cause melting there, requiring similar pressures of brucite breakdown
as in (1). However, crustal melts forming from sediments and eclogite are tonalitic to rhyolitic (Hermann and Rubatto, 2009;
Klemme et al., 2002), limiting the efficiency of Mg transport.

(3) metasomatise the slab-mantle interface as part of a melange (forming serpentine, chlorite, ±talc, amphiboles) at various
depths. These metasomatic/mixed materials could then be transported into the arc-magma source region – by diapirism, partial
melting or dehydration.

Studies of the relative stability of brucite and serpentine reveal that assemblages containing both minerals typically are stable
to temperatures of ~400–550°C at 2–6 GPa (Kempf et al., 2020; Lazar, 2020; Johannes, 1968), whereas serpentine on its own
can be stable to higher temperatures in the same pressure range (Ferrand, 2019; Ulmer and Trommsdorff, 1995; Wunder and
Schreyer, 1997). These stability ranges are compatible with brucite being preferentially mobilized at depths of ~100-150 km
within the lithospheric mantle of the slab in cold and intermediate subduction zones (Van Keken and Wilson, 2023). Similarly,
previosuly inferred P-T ranges of fluid release from the slab may also support preferential breakdown of brucite as a source of
fluids to arc magma source regions (Hacker, 2008; Chemia et al., 2015; Konrad-Schmolke et al., 2016). On the other hand,
even though brucite breakdown may produce a free aqueous fluid, it is currently uncertain if such a fluid would carry much
Mg (Codillo et al., 2022; Kessel et al., 2005; Spandler et al., 2007; Carter et al., 2015; Manning, 2004; Stalder et al., 2001).

It is also possible that none of the conditions required for brucite to induce Mg isotope anomalies in arc magmas are met. In
this case, serpentinite is unlikely to explain variations in arc lava Mg isotope variations and an alternative process is required.
For example, the $\delta^{26}$Mg observed in (differentiated) arc lavas may not be a magma source signal reflecting the composition of
the mantle wedge, but may instead be a result of magmatic fractionation in the crust of the upper plate (Brewer et al., 2018).
Such a process is testable through detailed Mg isotope measurements of arc magmatic rocks.

**Author contribution**: SGN, FK and HRM conceived the study. SGN and FK designed the mineral dissolution experiments
and MA carried them out. MA and SGN collected concentration data, FK collected TGA data, and PAEPS collected Mg
isotope data. SGN prepared the manuscript with contributions from all co-authors.

**Competing interests**: The authors declare that they have no conflict of interest.

**Acknowledgements:** This study was supported by funds from NSF grant EAR-1829546 to SGN and the Joint Initiative
Awards Fund from the Andrew W. Mellon Foundation to SGN and FK.

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
