# Peer review of "Magnesium isotope fractionation processes during seafloor serpentinization and implications for serpentinite subduction"

_EGUsphere, 2023_

## Author Response (AR1)

**We thank both reviewers for their constructive comments and suggestions. Below we outline these comments as well as our response to each comment, including a description of what changes were made in the manuscript to address the comment. Our responses are in red italics.**

**Reviewer 1:**

The paper by Nielsen et al. reported new data for serpentinite from a variety of oceanic settings. The authors found relatively homogenous mantle-like Mg isotopic compositions for bulk serpentinites, but significant heavy Mg isotope enrichment in brucite and iowaite within the serpentinites. Based on these results, the authors proposed that preferential breakdown of brucite/iowaite could release isotopically heavy Mg into the sub-arc mantle, which may explain the heavy-d26Mg observed in some arc volcanic rocks. In general, this is a neat study with some relatively significant results, which I enjoyed reading. I have some minor comments which mostly focus on the lack of information provided in the text which should be easily addressed (detailed below); hence I recommend that this is suitable for publication after minor revisions.

(1) As mentioned in this paper, a few studies have reported Mg isotope data for serpentinites. I would suggest the authors to make a comparison with the literature data visually through figures. It may also be worthwhile to draw a figure for comparison with arc magmas. *- we have added the literature serpentinite data field to the figure outlining our bulk serpentinite data (formerly figure 3, now figure 4). However, we are not convinced that drawing a new figure with arc lava Mg isotope data has substantial value since we do not present such data here and the comparison, therefore, is not direct.*

(2) The amount of fluid leached from the bulk serpentinite dissolution experiment is not shown. Please specify this amount and the method used to determine it." *- We do not understand what the reviewer means by 'fluid leached from the bulk serpentinite'. We did not leach fluid from the serpentinites, we dissolved brucite and iowaite, whereas serpentine remained in the residue. We used TGA measurements of the residues to verify that brucite and iowaite had been dissolved. Perhaps the reviewer would like to know the volume of acetic acid that was used to leach the serpentinites? We have supplied this information in the manuscript (Lines 133 and 177).*

Other comments

Line 150: I'll suggest to cite some method papers here as well (e.g., Teng et al., 2015 GGR). *- we have already cited the methods papers that most closely align with those used here. So we would prefer to cite these in order to avoid confusion about the detailed methodology.*

Line 203: "olivine" *- corrected*

Line 295: The format of (Li, 2023) is not correct. *- corrected*

Table 1: Please clarify how 2se was calculated and how n was defined in this study. *- we have added explanation at the bottom of table 1*

Reviewer 2:

Nielson et al. carried out a systematic Mg isotope study for a set of serpentinites formed both at mid-oceanic ridge settings and at forearc settings in subduction zones. They found that there is no significant Mg isotope variations of serpentinites during serpentinization of peridotite. They also performed partial dissolution experiments that dissolved mainly brucite and found that brucite is preferentially enriched in isotopically heavy Mg than serpentine. Based on these results, they discussed the implications of serpentinite subduction for the generation of arc magmas with heavy Mg isotope compositions. I find the topic is interesting and important for understanding Mg isotope systematics in subduction zones, especially for the formation of arc magmas with heavy Mg isotope compositions. The interpretations are mostly supported by the data, and the writing is concise and clear. However, I still have some concerns on the presentation and interpretations.

My comments are as follows.

Line 61-62: the fluids with heavy Mg isotopes were considered by Teng et al. (2016) to be slab derived, whether it is due to serpentine breakdown is not explicitly expressed. *- we have changed the reference to Hu et al 2020 that do explicitly mention this process*

Line 67-68: it is better to revise "these T and P" to "such P-T conditions" *- corrected*

Line 172-174: the serpentinites from Puerto Rico trench show a large variation of d26Mg (~0.1‰), and the high d26Mg ones were ascribed to chemical weathering. However, there is no further information on the high-d26Mg serpentinites. Do they indeed reflect high degrees of chemical weathering? Please discuss this in the context of petrology and other geochemical data. *- we have added some text and a new figure 1 that demonstrates the weathering processes experienced by the Puerto Rico trench samples. Lines 110-112, 201-202*

Fig. 3 Caption: please note that Mariana forearc serpentinites are not seafloor serpentinites. *- we have changed the description to oceanic serpentinites*

Lines 175-180: please explain the data of high-d26Mg serpentinites in Puerto Rico in detail. From the present discussion, we can not evaluate whether the high-d26Mg signature is caused by chemical weathering, or instead, through other process like serpentinization. I suggest to discuss this point by integrating with petrographic observations as well as bulk rock major and trace element compositions. In addition, an evaluation of the Mg mobility during serpentinization/weathering process may be helpful. *- we have added some text and a new figure that demonstrates the weathering processes experienced by the Puerto Rico trench samples. Lines 110-112, 201-202*

Line 200: the Mg isotope variation of Puerto Rico serpentinites is significant, ~0.1‰. Considering the high MgO contents of serpentinites, this Mg isotope variation is essential. *- we have added some text and a new figure that demonstrates the weathering processes experienced by the Puerto Rico trench samples. Lines 110-112, 201-202*

Line 203: Typo: Olivine *- corrected*

Line 204-206: Even we assume that the serpentinization process does not result in serpentinites with heavy Mg isotope compositions, the chemical weathering process does. Please consider the possibility that the subduction can also involve the weathered serpentinites. *- Weathering typically takes place only in the uppermost portion of the crust where cold seawater can interact with serpentinites. The total volume of weathered serpentinites, therefore, remains relatively small compared with overall serpentinite abundances in slabs where this lithology is sufficient to control the Mg budget. We have added a few sentences to make this point clearer. Lines 233-236*

Line 265-275: There is large debate on the Mg isotope fractionation between serpentine and brucite. I am curious that if there is abundant brucite in the serpentinite sample, why not separate it and directly measure it? Is it difficult to separate brucite from serpentine in serpentinite? On the other hand, the serpentine is easy to be separated and directly measured. Then, a mass balance can be used to

cross-check the fractionation trend and magnitude between serpentine and brucite. - *Neither serpentine nor brucite can be mechanically separated because they are typically intergrown and very fine grained (which we also noted in line 282 of the original manuscript). Had these minerals been sufficiently large and separate from each other then it would indeed have made sense separate them mechanically.*

Line 310-315: for the warm to hot subduction zones, the atg + brucite dehydration reaction occurs typically below ca. 2 GPa. Only at the extremely cold subduction zones, this reaction can occur at 3 GPa (See Kendrick et al., 2013-EPSL). Thus, it seems difficult to release fluids through this reaction at subarc depths in most subduction zones. - *We agree with the reviewer. In the original manuscript we already stated that the P-T conditions required for brucite to break down and become a source of Mg to arc magmas was only found in cold and intermediate subduction zones, which was supported by the most recent subduction zone thermal models (line 314 of original manuscript).*